# Detection and evolutionary characterization of arboviruses in mosquitoes and biting midges on Hainan Island, China, 2019–2023

Qun Wu[1,2,3☯], Dingwei Sun[3☯], Wahid Zaman[1,2], Fei Wang[1,2], Doudou Huang[1], Haixia Ma[1], Shunlong Wang[1,2], Ying Liu[3], Puyu Liu[3], Xuexia Zeng[3], Zhiming Yuan🔾[1,2]*, Han Xia🔾[1,2,4]*

1 Key Laboratory of Virology and Biosafety, Wuhan Institute of Virology, Chinese Academy of Sciences, Wuhan, Hubei, China, 2 University of Chinese Academy of Sciences, Beijing, China, 3 Hainan Provincial Center for Disease Control and Prevention, Haikou, Hainan, China, 4 Hubei Jiangxia Laboratory, Wuhan, Hubei, China

☯ These authors contributed equally to this work.
* yzm@wh.iov.cn (ZY); hanxia@wh.iov.cn (HX)

**Data Availability Statement:** The viral genomes sequenced in this study were deposited in Genbank (https://www.ncbi.nlm.nih.gov/genbank/) /Genbase (https://ngdc.cncb.ac.cn/genbase/) and the

## Abstract

We conducted a large-scale survey of arboviruses in mosquitoes and biting midges to assess the presence and spread of mosquito-borne pathogens currently circulating on Hainan Island, China. RT-PCR assays were used to detect the arbovirus species, distribution, and infection rates in mosquitoes and biting midges. Cell inoculation and high throughput sequencing were performed to isolate the viruses and assemble full viral genomes. Phylogenetic analysis was conducted to identify the viral genotypes and evolutionary relationships with known viruses. During 2019–2023, 32,632 mosquitoes and 21,000 biting midges were collected from 14 of 18 cities/counties on Hainan Island. Japanese encephalitis virus (JEV) was detected in *Culex* mosquitoes from five cities/counties, where the minimum infection rate (MIR) was 1.6 (0.6–2.6) per 1,000 females tested. Tembusu virus (TMUV) was detected in *Culex* mosquitoes from three cities/counties with MIR1.0 (0–2.2) per 1,000. Getah virus (GETV) was detected in *Armigeres* mosquitoes from Qionghai city with MIR 7.1 (0–15.2) per 1,000. Oya virus (OYAV) and Bluetongue virus (BTV) were detected in biting midges from Wanning city with MIRs of 0.4 (0–1.2) and 0.1 (0–10.2) per 1,000, respectively. Three JEV strains were isolated and clustered within the genotype I group, which is presently the dominant genotype in China. Three TMUV strains were isolated for the first time on Hainan Island that belonged to Cluster 3. Three isolated GETVs were identified as Group 3. BTV was reported for the first time on Hainan Island, and the complete genome for one BTV strain was successfully assembled, which was classified as serotype 1 based on the sequences of segment 2. These results stress the need to develop adequate surveillance plan measures to better control the public health threat of arboviruses carried by mosquitoes and biting midges in local regions.

accession number was available in Table 1. All other relevant data are in the manuscript and supporting information files.

**Funding:** This study was supported by the National Key Research and Development Program of China (2022YFC2302700) to Han Xia, National Natural Science Foundation of China (U22A20363) to Zhiming Yuan, Hainan Province Science and Technology Special Fund (ZDYF2023SHFZ134 and ZDKJ2021035) to Dingwei Sun, Youth Program of Wuhan Institute of Virology (2023QNTJ-03) to Han Xia, and the European Virus Archive Global (EVA-GLOBAL) project that has received funding from the European Union's Horizon 2020 research and innovation programme (No. 871029) to Zhiming Yuan.The funders had no role in study design, data collection and analysis, decision to publish, or preparation of the manuscript.

**Competing interests:** The authors have declared that no competing interests exist.

## Author summary

In the early 1990s, a large-scale survey for arboviruses in vectors was conducted in Hainan Island. However, in the past 30 years, with climate change, urbanization, and the increasing population and trades, whether the arboviruses spectrum changed on this Island is unclear. We screened arboviruses in mosquitoes and biting midges from 2019 to 2023 to assess the presence of arbovirus and the genetic evolution currently circulating on this Island. Except for arboviruses such as JEV, GTEV, and OYAV, which have been previously reported circulating on Hainan Island, TMUV and BTV were detected for the first time during this survey. These findings highlight the increasing risk of arbovirus transmission in Hainan Island. It is crucial to prioritize extensive and continuous surveillance of arboviruses carried by mosquitoes and biting midges in local regions to help prevent the spread of emerging and re-emerging arboviral diseases.

## Introduction

The World Health Organization reported that vector-borne diseases account for more than 17% of all infectious diseases worldwide and cause more than 700,000 deaths annually. More than 3.9 billion people in over 129 countries are at risk of contracting dengue fever, with an estimated 96 million symptomatic cases and 40,000 deaths every year [1]. At least 530 viruses are listed in the U.S. Centers for Disease Control and Prevention arbovirus catalog (https://wwwn.cdc.gov/arbocat/), around 134 of which are known to cause human disease [2].

Mosquitoes and midges are important vectors of arboviruses that infect humans or animals. *Aedes* mosquitoes are major vectors of diseases caused by dengue virus (DENV), chikungunya virus (CHIKV), yellow fever virus, and Zika virus. In addition, *Culex* mosquitoes transmit viruses such as Japanese encephalitis virus (JEV) and West Nile virus [3]. More than 50 viruses have been isolated from *Culicoides* spp. worldwide. Most of the viruses belong to the families of *Sedoreoviridae*. (e.g., African horse sickness virus (AHSV) and bluetongue virus (BTV)), *Rhabdoviridae* (e.g., bovine ephemeral fever virus) and *Peribunyaviridae* (e.g., Akabane virus, Schmallenberg virus, or Oropouche virus). Among them, AHSV and BTV cause Category A animal disease, which was notified by the World Organization for Animal Health [4–6].

Hainan Island, located in the tropical and subtropical zones is separated from mainland China by the Qiongzhou Strait, and it comprises a unique and independent natural ecosystem with a typical tropical marine monsoon climate. These conditions are conducive to the survival and reproduction of various arthropod vectors, including mosquitoes, ticks, and biting midges, that facilitate the circulation of arboviruses among vectors and vertebrates [7,8]. Hainan Island is an endemic region for dengue, with outbreaks occurred in 1979, 1980–1991, and 2019, and for Japanese encephalitis (JE) with outbreaks also happened during 1952–1990 [9–11]. Arbovirus surveillance efforts conducted among vectors on Hainan Island during the 1960s to the 1990s detected and isolated viruses such as DENV, JEV, CHIKV, and Ross River viruses from mosquitoes, ticks, and bats [12]. However, rapid economic development as well as increased tourism and urbanization have significantly altered the natural ecological environment on Hainan Island and the distribution of vectors. Despite these changes, no large-scale systematic investigations of arboviruses in mosquitoes and biting midges have been conducted on the island since the 1990s until in 2017–2018 [13]. Thus, in the present study, we aimed to assess the arboviruses circulating in mosquitoes and biting midges on Hainan Island based on a large-scale field survey.

## Methods

### Mosquitoes and biting midges sampling

Human-baited double net for collecting anthropophagic day-biting *Aedes* and lamp trapping or animal tent trapping techniques for night-biting mosquitoes or biting midges were used. Mosquitoes were collected from March to October, and biting midges were captured from April to August. Mosquitoes and biting midges were collected from 14 out of 18 cities/counties on Hainan Island between 2019 and 2023 (Fig 1), covering the island's east, west, south, north, and central regions. The specimens were frozen immediately after capturing using dry ice at the collection site, and then transported to the vector laboratory of Hainan Provincial Center for Disease Control and Prevention (CDC) through cold chain with dry ice. The mosquito/midge morphological identification, sample grouping and homogenization, and RNA

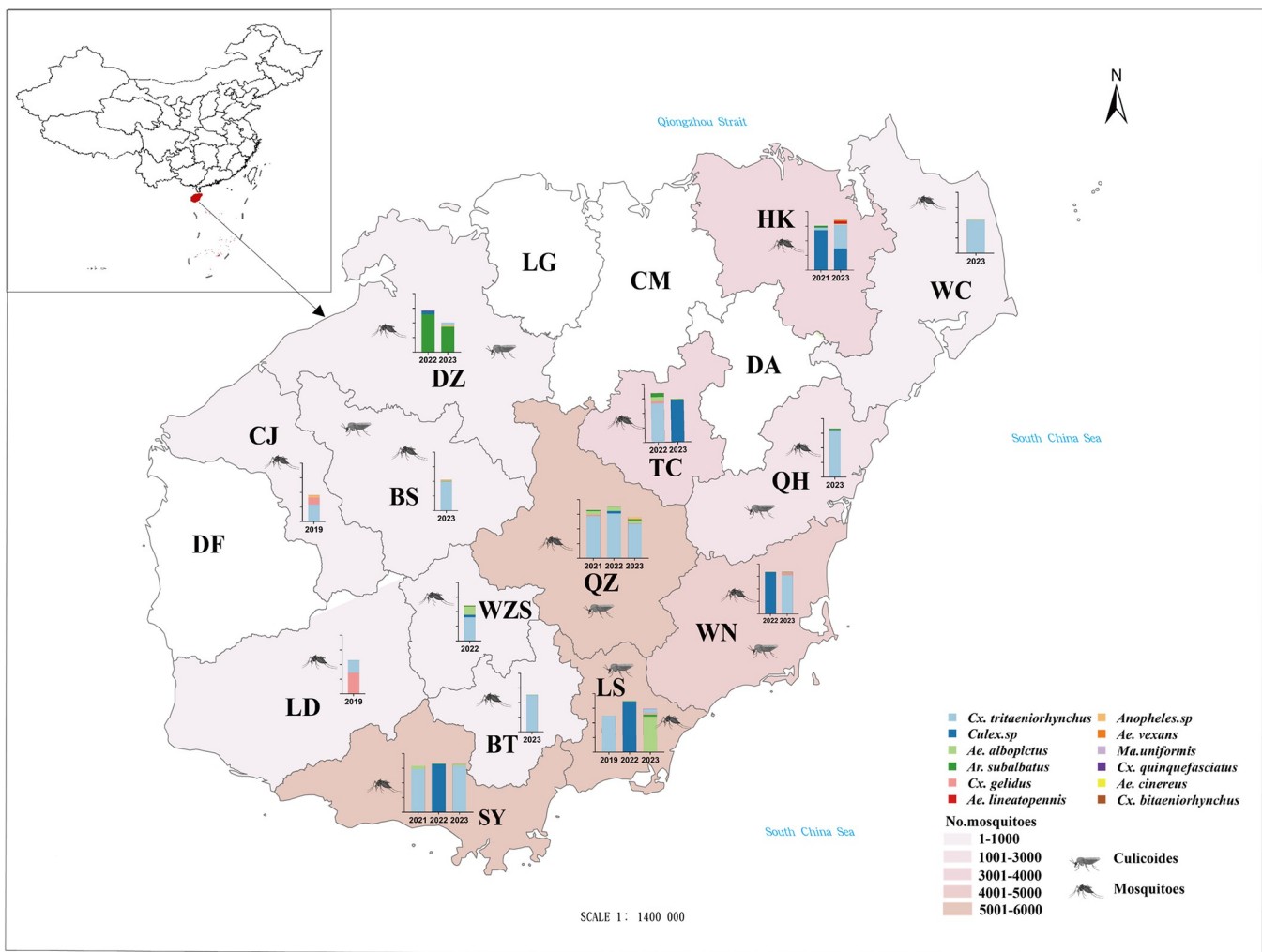

**Fig 1. Sampling locations for mosquitoes and *Culicoides* biting midges between 2019 and 2023 on Hainan Island, China.** HK: Haikou city, WC: Wenchang city, QH: Qionghai city, WN: Wanning city, LS: Lingshui county, TC: Tunchang county, QZ: Qiongzhong county, BT: Baoting county, SY: Sanya city, DZ: Danzhou city, BS: Baisha county, CJ: Changjiang county, LD: Ledong county, LG: Lingao country, CM: Chengmai country, DA: Dingan country, DF: Dongfang city. (Note: Blank cities refer to areas that have not been sampled. In the stacked bars, the left Y-axis represents the Log10 values of collected mosquitoes, with different colors indicating various mosquito species. The source of the basemap shapefile was from Resource and Environmental Science Data Platform (https://www.resdc.cn/DOI/doi.aspx?DOIid=121) [19]. The visualization for the map was constructed through R v4.3.3 (https://www.r-project.org/) package ggplot2 3.5.1(https://ggplot2.tidyverse.org/).The images in Fig 1 created in BioRender. Huang, D. (2024) BioRender.com/u66x097).

detection were conducted in Hainan Provincial CDC. Female adult mosquitoes were identified at the species level and pooled (30–60 mosquitoes/pool) by collection site, date, and species. We identified biting midges only to genus level (*Culicoides*), with 300–400 midges/pool. A portion of the homogenization supernatants for the sample were transported to the Wuhan Institute of Virology, Chinese Academy of Sciences, through cold chain with dry ice and then maintained in a refrigerated container for virus isolation in a biosafety laboratory.

## Sample preparation and RNA extraction

Mosquitoes and biting midges were assigned to different pools according to the collection site, species, and gender. Next, 1,000 μL of Roswell Park Memorial Institute (RPMI) medium was added to each pool, then triturated with the cryogenic grinding method by using a High-Speed Low-Temperature Tissue Grinding Machine with two 30-s cycles at 50 Hz. After sufficient grinding, macerated mosquitoes and biting midges were centrifuged at $15,000 \times g$ (4˚C for 30 min) to remove any cell debris and bacteria [14]. Supernatants were stored at −80˚C. RNA was extracted from 200 μL of each homogenized supernatant sample with a 600 μL Trizol and an RNA extraction kit according to the manufacturer's instructions.

## Arboviruses detection through RT-PCR

Mosquitoes and biting midges extracts were screened for arboviruses by RT-PCR according to previously described methods [13,15,16]. Briefly, complementary DNA (cDNA) was synthesized using a cDNA synthesis kit. The reaction conditions were 95˚C for 2 min, followed by 34 cycles at 95˚C for 15 s, 52˚C to 60˚C (see S1 Table) for 15 s, and 72˚C for 15 s; and then 72˚C for 5 min, before reducing the temperature to 4˚C until the product was removed from the thermocycler. The primer sequences and characteristics are shown in S1 Table. The minimum infection rate (MIR) was calculated as (the number of positive pools/total number of individual mosquitoes or midges tested) × 1,000, and 95% confidence intervals were determined using the chi-square calculator in SPSS version 21 (International Business Machines Corporation).

## Arboviruses isolation and purification

All homogenates generated in the study were inoculated onto C6/36 (*Aedes albopictus* RNAi-deficient), and BHK-21 (baby hamster kidney) cells, and Vero-E6 cells for virus isolation. Viruses were isolated as described in a previous study [14]. Briefly, the remaining parts of the mosquito or midge homogenates were removed from storage at -80˚C and kept on ice. Micro-filtered supernatants (100 μL) from each pool were inoculated into each well of 24-well plates containing C6/36 cells as passage one. After 1 h of incubation (C6/36) at 28˚C, the inoculum was removed and replaced with RPMI (with 2% fetal bovine serum (FBS) and 2% (penicillin-streptomycin-amphotericin B (PSA) solution, and the cell plates were placed in the incubator at 28˚C with 5% $CO_2$ for 7 days, followed by two passages in C6/36 cells. The remaining mosquito or midge homogenates also were inoculated into BHK-21 and Vero-E6 cells according to the same steps described above, but with an incubation temperature of 37˚C and using cell maintenance fluid comprising Dulbecco's modified Eagle medium with 2% FBS and 2% PSA. The supernatants (100 μL) from the third passage in C6/36 cells were seeded onto BHK-21 cell plates (24-well plates) and incubated for 7 days to monitor cytopathic effects (CPEs).

## Sequencing and phylogenetic analysis

Envelope genes in JEV fragments were amplified from cDNA with positive pools. The primers were as follows [17]: JEV-E-F, 5′-TGYTGGTCGCTCCGGCTTA-3′; and JEV-E-R,

5′-GATGTCAATGGCACATCCAGT-3′. The amplicons were purified with a Gel Extraction Kit and each cDNA was cloned into the pCE3 Blunt Vector and transformed into DH5α Competent Cells. Sequencing libraries were prepared with a RNA-seq Library Prep Kit for Illumina according to the manufacturer's instructions. Whole genomes were sequenced by next-generation sequencing, and phylogenetic analyses were performed as described previously [14]. The sequences were identified by BLAST searches using GenBank. The sequences obtained were aligned with MegAlign, and neighbor-joining trees were generated with MEGA v. 11 using 1,000 bootstrap replicates. Phylogenetic trees were visualized, modified, and annotated by using Tree Visualization by One Table (tvBOT; https://www.chiplot.online/normalTree.html) [18].

## Results

### Mosquito and biting midge samples

During 2019 to 2023, we collected 32,632 mosquitoes from 14 cities/counties on Hainan Island. The mosquitoes consisted of *Culex tritaeniorhynchus* (35.0%), *Ae. albopictus* (16.5%), *Armigeres subalbatus* (9.8%), *Culex gelidus* (5.3%), and unidentified *Culex* spp. (contained *Cx. tritaeniorhynchus* and *Cx. gelidus*, because some dorsal scales of *Cx. gelidus* were missing, making them morphologically indistinguishable from *Cx. tritaeniorhynchus*) (26.7%) and other species in limited numbers (6.7%) (Fig 1 and S2 Table). Approximately 21,000 *Culicoides* biting midges were also collected from six cities/counties (Qiongzhong, Danzhou, Lingshui, Wanning, Qionghai, and Baisha) (Fig 1 and S2 Table).

### Detection of arboviruses in mosquitoes and biting midges

Seventeen pools tested positive by RT-PCR, including 15 mosquito pools: nine for JEV, three for Tembusu virus (TMUV), and three for Getah virus (GETV). Additionally, two biting midge pools were positive, one for the Oya virus (OYAV) and one for the BTV (Table 1). In particular, JEV was detected in *Culex* spp. from mosquitoes in Wanning on the eastern coast of Hainan Island in 2022 and *Cx. tritaeniorhynchus* mosquitoes from four cities/counties (Haikou, Qionghai, Qiongzhong, and Baoting) during 2023 covering the northern, eastern coastal, central, and southeastern regions of Hainan Island, respectively. The collection specific MIRs ranged from 0.8 (0–2.3) to 4.6 (0–13.5) per 1,000. In addition, TMUV was detected in *Cx. gelidus* in Wanning, *Cx. tritaeniorhynchus* in Sanya, and *Culex* spp. in Tunchang during 2023, with MIRs of 1.4 (0–4.2), 0.7 (0–2.1), and 1.4 (0–4.0) per 1,000, respectively. Furthermore, GETV was detected in *Ar. subalbatus* from Qionghai with an MIR of 7.1 (0–15.2) per 1,000 (Table 1). OYAV was detected in biting midges from Waning in 2022 with an MIR of 0.4 (0–1.2) per 1,000, and BTV was detected in biting midges from Qiongzhong in 2023 with an MIR of 0.1 (0–10.2) per 1,000. Due to inconsistencies in sampling time and location across collection sites and human-baited double net with low efficiency for capturing *Aedes* mosquitoes [20], arbovirus risk estimates can only reflect site-specific risks rather than an overall risk.

### Viruses' isolation and phylogenetic analyses

In total, 765 pools of mosquito and biting midge homogenates were inoculated onto C6/36 cells, BHK-21 cells, and Vero-E6 cells, and a total of nine viral isolates were obtained, of which eight strains were able to cause CPEs in C6/36 cells, and one strain (HN-WN22-Cu-18) caused CPEs in BHK-21 cells. Two of the viral isolates observed CPEs in C6/36 cells(HN-SY23-Ct-05 and HN-WN23-Cg-02)also caused CPEs in BHK-21 cells, and one (HN-QH23-As-10) caused CPEs in Vero-E6 cells. Five strains(HN-QH23-Ct-01, HN-QH23-Ct-13, HN-QH23-As-12,

**Table 1. Arboviruses identified in mosquitoes and biting midges on Hainan Island, 2019–2023.**

| Virus | Strain | Location | Year | Mosquito/ biting midge sample genus/ species | Habitant | PCR Screening | | | | | Isolation | Accession No. | Storage No. |
|---|---|---|---|---|---|---|---|---|---|---|---|---|---|
| | | | | | | Positive Pools | No. pools | Positive pool rate (%) | NO. individuals | MIR (‰) 95% CI | | | |
| Janpanese encephalitis virus (JEV) | HN-WN22-Cu-18 | WN | 2022 | *Culex* spp. | Cattle farm | 3 | 24 | 12.5 (0–26.8) | 2242 | 1.3 (0–2.9) | YES | PP682372[a] | CSTR: 16533.06. IVCAS 6.9360 |
| | HN-WN22-Cu-04 | | | | | | | | | | NO | NA | NA |
| | HN-WN22-Cu14 | | | | | | | | | | NO | NA | NA |
| | HN-QH23-Ct-01 | QH | 2023 | *Cx. tritaeniorhynchus* | Cattle farm | 3 | 19 | 15.8 (0–33.9) | 1650 | 1.8 (0–3.9) | YES | PP682373[a] | CSTR: 16533.06. IVCAS 6.9361 |
| | HN-QH23-Ct-13 | | | | | | | | | | YES | PP682374[a] | CSTR: 16533.06. IVCAS 6.9362 |
| | HN-QH23-Ct-18 | | | | | | | | | | NO | NA | |
| | HN-HK23-Ct-20 | HK | 2023 | *Cx. tritaeniorhynchus* | Cattle farm Rice paddy | 1 | 22 | 4.6 (0–14.00) | 1284 | 0.8 (0–2.3) | NO | NA | NA |
| | HN-QZ23-Ct-01 | QZ | 2023 | *Cx. tritaeniorhynchus* | Cattle farm | 1 | 4 | 25.0 (0–100) | 220 | 4.6 (0–13.5) | NO | NA | NA |
| | HN-BT23-Ct-03 | BT | 2023 | *Cx. tritaeniorhynchus* | Cattle farm | 1 | 4 | 25.0 (0–100) | 330 | 3.0 (0–9.0) | NO | NA | NA |
| Tembusu virus (TMUV) | HN-SY23-Ct-05 | SY | 2023 | *Cx. tritaeniorhynchus* | Cattle farm | 1 | 21 | 4.8 (0–14.7) | 1448 | 0.7 (0–2.1) | YES | PP682375[a] | CSTR: 16533.06. IVCAS 6.9363 |
| | HN-TC23-Cu-03 | TC | 2023 | *Culex* spp. | Rubber Grove ,Cattle | 1 | 8 | 12.5 (0–42.1) | 742 | 1.4 (0–4.0) | YES | PP682376[a] | CSTR: 16533.06. IVCAS 6.9364 |
| | HN-WN23-Cg-02 | WN | 2023 | *Cx. gelidus* | Cattle farm | 1 | 12 | 8.3 \(0–26.7) | 714 | 1.4 (0–4.2) | YES | PP682377[a] | CSTR: 16533.06. IVCAS 6.9365 |
| Getah virus (GETV), | HN-QH23-As-10 | QH | 2023 | *Ar. subalbatus* | Cattle farm | 3 | 14 | 21.4 (0–46.0) | 420 | 7.1 (0–15.2) | YES | PP682378[a] | CSTR: 16533.06. IVCAS 6.9366 |
| | HN-QH23-As-12 | | | | | | | | | | YES | PP682379[a] | CSTR: 16533.06. IVCAS 6.9367 |
| | HN-QH23-As-14 | | | | | | | | | | YES | PP682380[a] | CSTR: 16533.06. IVCAS 6.9368 |
| Oya virus (OYAV) | HN-WN22-Cul-04 | WN | 2022 | *Culicoides* spp. | Cattle farm | 1 | 8 | 12.50 (0–42.06) | 2400 | 0.42 (0–1.23) | NO | NA | NA |

*(Continued)*

**Table 1.** (Continued)

| Virus | Strain | Location | Year | Mosquito/ biting midge sample genus/ species | Habitant | PCR Screening | | | | | Isolation | Accession No. | Storage No. |
|---|---|---|---|---|---|---|---|---|---|---|---|---|---|
| | | | | | | Positive Pools | No. pools | Positive pool rate (%) | NO. individuals | MIR (‰) 95% CI | | | |
| Bluetongue virus (BTV) | HN-QZ23-Cul-12 | QZ | 2023 | *Culicoides* spp. | Cattle farm | 1 | 39 | 2.56 (0–7.75) | 13650 | 0.07 (0–0.22) | NO | C_AA084017.1[b] C_AA084019.1[b] C_AA084020.1[b] C_AA084021.1[b] C_AA084022.1[b] C_AA084023.1[b] C_AA084024.1[b] C_AA084025.1[b] C_AA084018.1[b] | NA |

Note: HK: Haikou city, WC: Wenchang city, QH: Qionghai city, WN: Wanning city, LS: Lingshui county, TC: Tunchang county, QZ: Qiongzhong county, BT: Baoting county, SY: Sanya city, DZ: Danzhou city, BS: Baisha county, CJ: Changjiang county, LD: Ledong county. (Note: "a" was the GenBank accession number and "b" was the Genbase accession number).

HN-QH23-As-14, and HN-TC23-Cu-03)were inoculated onto C6/36 cells for three generations, and then the supernatants were used to inoculate BHK-21 cells, resulting in the development of CPEs in the BHK-21 cells.

Based on the nucleotide sequence of the whole genome, JEV is currently classified into five different genotypes: GI, GII, GIII, GIV, and GV (Fig 2A) [21,22]. Three strains of JEV were successfully isolated. One strain (HN-WN22-Cu-18) was isolated from *Culex* spp. in Wanning during 2022, and the other two (HN-QH23-Ct-01 and HN-QH23-Ct-03) were from *Cx. tritaeniorhynchus* in Qionghai during 2023. These isolates clustered together within GI (Fig 2A), which is the dominant genotype in China at present. Phylogenetic analysis based on the whole-genome of JEV showed that HN-WN22-Cu-18 and HN-QH23-Ct-01 clustered with two strains identified in mosquito samples collected during 2017 from Guangxi and Guangdong provinces in China, where HN-QH23-Ct-13 was more closely related to the samples obtained from *Cx. tritaeniorhynchus* collected during 2015 in Jiangsu province in China (Fig 2A).

The E protein of JEV is considered the main site for host–virus attachment and it consists of three structural domains: domain I (DI; E1–E51, E137–E196, and E293–E311), domain II (DII; E52–E137 and E197–E292), and domain III (DIII; E310–E411) [24–26]. The deduced amino acid differences in E protein sequences were aligned for comparison among the newly sequenced strains, strains detected in mosquitoes from Hainan Island, SA14 strain (KU323483.1), and the vaccine strains (SA14-14-2 and AF495589.1) currently used in China (S3 Table). Fourteen amino acid residues in the newly detected Hainan JEV strains differed from those in the live attenuated vaccine SA14-14-2-derived strain: three in DI with E138 (Lys→Glu/Gln), E176(Val→Ile), and E177(Ala →Thr); six in DII with E107(Phe→Leu), E129 (Thr→Met), E222(Ala→Ser), E244(Gly→Glu/Gln), E264(His→Gln), and E279(Met→Lys); three in DIII with E315(Val→Ala), E327(Ser→Thr), and E366(Ala→Ser); and two outside the domains with E439(Arg→Lys) and E447(Gly→Val). Five mutation sites differed between the newly detected Hainan JEV strains and the SA14 strain comprising: E129(Thr→Met), E222 (Ala→Ser), E327(Ser→Thr), E366(Ala→Ser), and E447(Gly→Val).

We did not successfully obtain the BTV isolate, but the complete genome of BTV HN-QZ23-Cul-12 obtained from biting midges in Qiongzhong during 2023 was successfully assembled. Phylogenetic analysis based on segment 2 (VP2) showed that this Hainan BTV strain belonged to serotype 1 and it was closely related to BTVs isolated from India and Greece (Fig 2B) with similarities of 95.66% and 94.50%, respectively.

Three strains of TMUV (HN-TC23-Cu-03, HN-WN23-Cg-02, and HN-SY23-Ct-05) were isolated for the first time in Hainan Island (Table 1) and assigned to Cluster 3, They were closely related to strains identified in mosquito samples collected during 2012 and 2020 in Yunnan (Fig 2C).

In addition, three GETV isolates (HN-QH23-As-10, HN-QH23-As-12, and HN-QH23-As-14) isolated from *Ar. subalbatus* during 2023 in Qionghai were identified as G3, and they clustered into a separate clade with the closest evolutionary distance to strain HND1712-1 isolated from *Cx. tritaeniorhynchus* during 2017 in Hainan (Fig 2D). The deduced amino acid sequences of envelope protein 2 were aligned to compare the newly isolated GETV strains (HN-QH23-As-10, HN-QH23-As-12, and HN-QH23-As-14) with the M1 strain and HND1712-1 strain isolated previously in Hainan. Nine mutation sites differed between the GETV strains (HN-QH23-As-10, HN-QH23-As-12, and HN-QH23-As-14) and the M1 strain (S4 Table), i.e., E109(Gly→Asp), E134(Ala→Thr), E205(Arg→Ser), E269(Leu→Val), E323 (Asp→Glu/Gln), E368(Val→Ala), E374(Cys→Gly), and E378(Val→Ile). Two mutation sites comprising E134(Ala→Thr) and E283(Le→Thr) differed compared with the HND1712-1 strain. Thus, the GETV strains present in the Hainan area were quite different from the M1 strain isolated in 1964, but no additional frequent mutations had appeared in recent years.

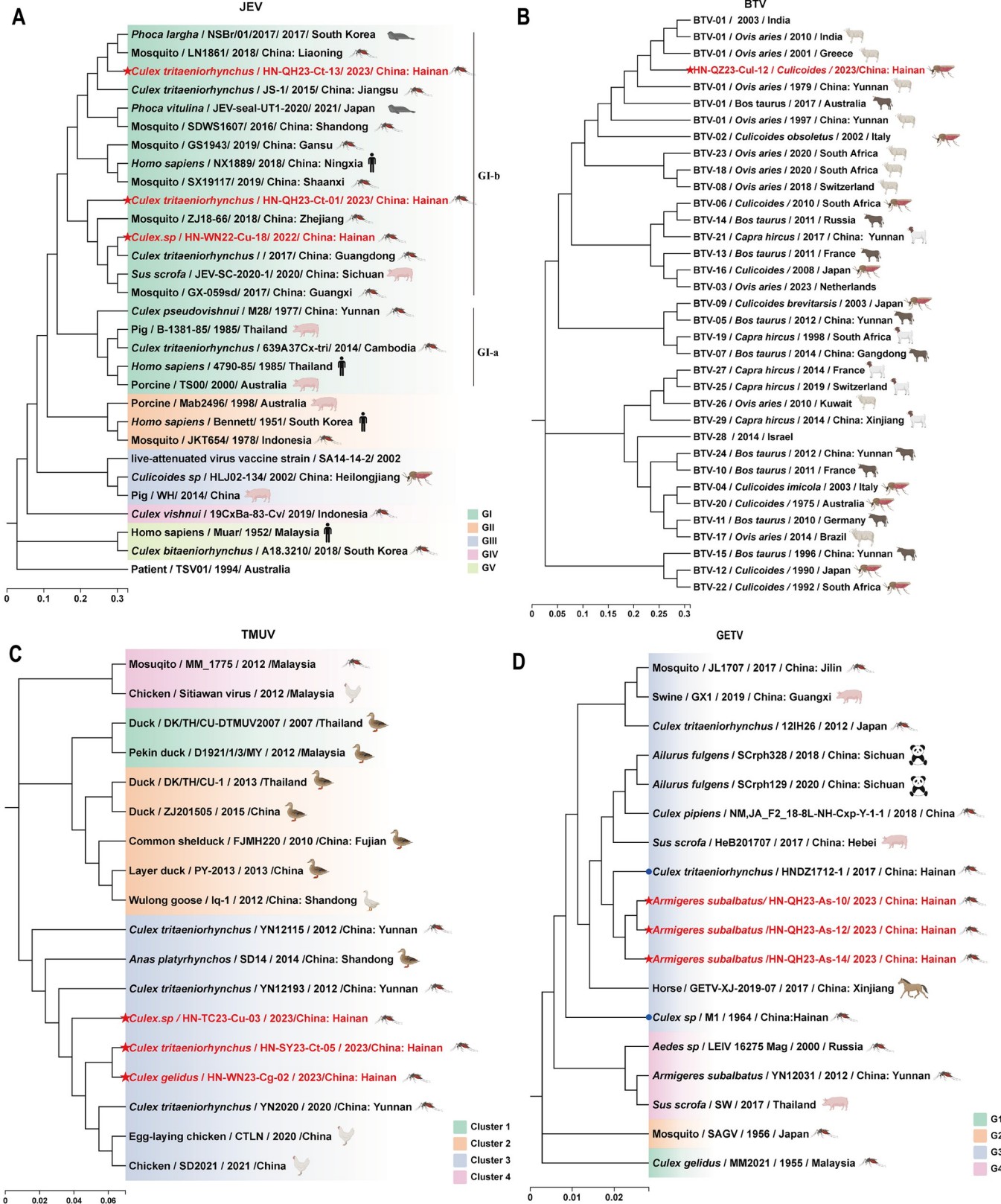

**Fig 2. Phylogenetic analysis of newly identified arboviruses isolated from mosquitoes/midges with complete genomes in Hainan Island during 2019–2023.** (A) Phylogenetic tree constructed based on the complete genome of Japanese encephalitis virus (JEV). (B) Phylogenetic tree constructed based on the

complete sequence of segment 2 in Bluetongue virus (BTV). (C) Phylogenetic tree constructed based on the complete genome of Tembusu virus (TMUV). (D) Phylogenetic tree constructed based on the complete genome of Getah virus (GETV). The blue circle indicates the viral strain reported previously in Hainan Island, and red highlights indicate the viral strains obtained in the present study. Sequences information in S5 Table. The images in Fig 2 were created in BioRender. Huang, D. (2024) BioRender.com/s17t999.

## Discussion

To strengthen epidemiological investigations of arboviruses and meet the need for early disease warnings, RT-PCR detection, virus isolation, and deep sequencing were used in combination in our study. JEV was isolated previously during 1983–1985 in Hainan [27]. However, there was no full genome sequence information and there were no further detections of JEV in mosquitoes from Hainan until 30 years later in 2017 [13]. The detection of JEV in *Cx. tritaeniorhynchus* and *Culex* spp. (contained *Cx. tritaeniorhynchus* and *Cx. gelidus*) pools from five cities and counties in Hainan Island confirmed that *Cx. tritaeniorhynchus* was the main vector of JEV, and it was widespread in Hainan. The phylogenetic tree showed that all strains belonged to GI in the same manner as those tested in 2017–2018 [13], and they were closely related to strains collected during 2015 and 2017 from Jiangsu, Guangxi, and Guangdong provinces in China (Fig 2A). The JEV isolates from mosquitoes, pigs, birds, and human belonged to GIII before 2008, and GI was the only isolated subsequently from mosquitoes [28]; However, in the last 15 years, JEV GI has replaced GIII as the dominant genotype in China [23,29]. Our results indicated that JEV GI is also the dominant genotype in Hainan Island. Further investigations are needed to identify whether other JEV genotypes might be present in Hainan Island.

The E protein is the crucial virulence determinant for JEV [30], and eight substitutions (E107, E138, E176, E177, E264, E279, E315, and E439) in the E protein gene affect the critical amino acids related to virus attenuation. Thus, they are often used as the critical targets for quality control in vaccine strains [25,31]. In our study, the newly detected Hainan JEV strains differed completely from the vaccine strains (SA14-14-2) in terms of the eight substitutions (S3 Table), but they were the same as those in the SA14 strain, thereby indicating that the JEV virulence has not changed dramatically in Hainan Island. Another five amino acid residues also differed (E129Thr→Met, E222Ala→Ser, E327Ser→Thr, E366Ala→Ser, and E447Gly→Val), and some or all were detected in previous studies in Shanghai province (E222, E327, and E366) and Shandong province (E129, E222, E327, and E366) [32–34]. In addition to the critical amino acid mutations, other mutations such as those at E129, E222, E327, E366, and E447 might cause immune escape from the vaccine, and thus further research is needed.

The first BTV isolate in China was found during 1979 in Yunnan province [35], and BTV infections in cattle were investigated subsequently in about 23 Chinese provinces [36]. There have been no previous reports of BTV detection or isolation on Hainan Island. In the present study, strain HN-QZ23-Cul-12 was detected and identified as serotype 1 (Fig 2B). Midges (*Culicoides* spp.) play an important role in the transmission of BTV [37]. BTV can be transmitted and circulate continuously between cattle and midges. Thus, cattle at the Qiongzhong collection site may have been infected with BTV, highlighting the need for increased monitoring of infections in local animals.

TMUV is an emerging mosquito-borne flavivirus, and it was previously identified in mosquitoes in Shandong [38] and Yunnan [39] provinces in China. Phylogenetic analyses of the whole genomes showed that TMUVs could be divided into four clusters [40]. Cluster 1, Cluster 2, and Cluster 4 mainly contain strains isolated from ducks, geese and chickens, and contribute to the prevalence of disease, e.g., strain D1921/1/3/MY caused outbreaks of neurological disease characterized by progressive paralysis in broiler duck flocks aged between 4 and 7 weeks

old in some areas of Malaysia [41]. Cluster 3 is considered a novel cluster of TMUVs, which contains strains originating from mosquitoes that have acquired attenuated pathogenicity compared with those in other clusters, especially Cluster 2 [42]. There have been no previous reports of TMUV isolates from mosquitoes collected on Hainan Island. The three Hainan TMUV strains were grouped within Cluster 3 based on the whole-genome tree, and these were closely related to strains identified in mosquitoes and chickens (Fig 2C). The results showed that the TMUV strains from Hainan were similar to other TMUV strains that originated in mosquitoes and chickens and that mosquitoes may have played a significant role in the spread of TMUV among local chickens in Hainan Island. A previous study found surprisingly high rates of seropositivity and duck Tembusu virus (DTMUV) positivity by RT-PCR in duck farm workers in China. Some people without contact with ducks also had high neutralizing anti-body titers to DTMUV in Thailand [43,44]. Thus, strengthening TMUV screening in mosquitoes and chickens is important, but special attention should also be paid to health monitoring for personnel involved in poultry farming in Hainan.

GETV is a neglected mosquito-borne alphavirus that causes pyrexia, body rash, and leg edema in horses, and fetal death and reproductive disorders in pigs [45]. GETVs have been isolated from 17 different mosquito species belonging to five genera in *Culicidae* (*Culex, Anopheles, Armigeres, Aedes*, and *Mansonia)*, as well as from midges in Eurasia. In China the GETV have been found in 22 of 34 provinces by March 2022 [46]. The isolation of GETVs on Hainan Island was expected as GETV was isolated in this area in 1964 [47] and reports of GETVs reappeared about 50 years later in 2017–2018 [13]. Phylogenetic analyses of whole genomes showed that the newly isolated Hainan GETV strains (HN-QH23-As-10, HN-QH23-As-12, and HN-QH23-As-14) were distant from the M1 strains isolated previously in 1964 but close to the HND1712-1 strain isolated during 2017 in Hainan (Fig 2D), presumably because the HND1712-1, HN-QH23-As-10, HN-QH23-As-12, and HN-QH23-As-14 strains and M1 strains had different origins, but the reasons for these differences need to be investigated further. The newly isolated Hainan GETV strains were assigned to G3 in the whole-genome tree (Fig 2D). GETV G3 is the leading group that causes animal diseases, and these viruses have caused multiple disease outbreaks in horses and pigs, and disease in cattle and blue foxes in recent years [45,46,48]. All of the new GETV strains were isolated from *Ar. subalbatus* mosquitoes, but more evidence is needed to confirm whether the *Ar. subalbatus* mosquito is a competent vector for GETV on Hainan Island.

Data from the Hainan Provincial CDC indicates that human cases of JE have rarely been reported on Hainan Island in recent years due to JE vaccination programs. From 1991 to 2018, Hainan reported few or no local dengue cases for over 20 years. However, since September 2019, the number of reported for local dengue cases has increased annually. The animal diseases associated with GETV, TMUV, and BTV have been endemic in mainland China, but there are currently no reports of this animal disease outbreak in Hainan. Our detection of JEV, GETV, TMUV, and BTV in vectors suggests an alarming increase in arbovirus diversity and a heightened risk of infection on Hainan Island. However, the lack of reported disease may be due to a lack of recognition, rather than due to the absence of disease. DENV was not found in this study is likely due to the limited number of *Ae. aegypti* and *Ae. albopictus* collected, where most of the *Ae. albopictus* samples were collected from Sanya and Qiongzhong, with lower numbers collected in other areas. Furthermore, recent dengue outbreaks in Hainan were primarily linked to imported cases. Thus, there may be a low prevalence of DENVs carried by local *Aedes* mosquitoes on Hainan Island at present. Sample size and coverage for *Aedes* mosquito surveillance should be increased.

Due to the interruptions caused by the COVID-19 outbreak, only Sanya and Qiongzhong were surveyed continuously for 3 years, while the other sampling sites were monitored for

only one or two years. Because of the inconsistencies in sampling time and location across collection sites, the data collected may not be well suited to distinguishing fluctuations in epidemic or zoonotic risk during the period of observation, our data still indicate a case could be made for using a One Health approach to monitoring arbovirus activity on Hainan Island in a more consistent manner to determine outbreak risk.

## Supporting information

**S1 Table. Primers used in heminested RT-PCR or RT-PCR assays for detecting arboviruses.**
(DOCX)

**S2 Table. Geographic and species information for samples collected on Hainan Island during 2019–2023.**
(DOCX)

**S3 Table. Amino acid mutation analysis for JEV envelope protein (E) protein detected on Hainan Island.**
(DOCX)

**S4 Table. Amino acid mutation analysis for GETV envelope protein 2 protein detected on Hainan Island.**
(DOCX)

**S5 Table. Sequences information in Fig 2.**
(DOCX)

## Acknowledgments

The authors would like to thank the Haikou, Wenchang, Qionghai, Wanning, Lingshui, Sanya, Tunchang, Qiongzhong, Wuzhishan, Baoting, Danzhou, Baisha, Changjiang, and Ledong Centers for Disease Control and Prevention, who provided help with mosquitoes and biting midge sample collection. We thank the assistance provided by Institutional Center for Shared Technologies and Facilities, and National Virus Resource Center of Wuhan Institute of Virology, CAS. We would like to thank Dr. James W. LeDuc's valuable suggestions.

## Author Contributions

**Conceptualization:** Zhiming Yuan, Han Xia.

**Data curation:** Qun Wu, Dingwei Sun, Fei Wang, Shunlong Wang.

**Formal analysis:** Fei Wang, Zhiming Yuan, Han Xia.

**Funding acquisition:** Dingwei Sun, Zhiming Yuan, Han Xia.

**Investigation:** Qun Wu, Dingwei Sun, Ying Liu, Puyu Liu, Xuexia Zeng.

**Methodology:** Qun Wu, Dingwei Sun, Doudou Huang, Haixia Ma, Han Xia.

**Project administration:** Zhiming Yuan, Han Xia.

**Resources:** Qun Wu, Dingwei Sun, Doudou Huang, Haixia Ma, Ying Liu, Puyu Liu, Xuexia Zeng.

**Supervision:** Zhiming Yuan, Han Xia.

**Validation:** Zhiming Yuan, Han Xia.

**Visualization:** Qun Wu, Dingwei Sun, Wahid Zaman, Han Xia.

**Writing – original draft:** Qun Wu, Dingwei Sun, Han Xia.

**Writing – review & editing:** Qun Wu, Dingwei Sun, Wahid Zaman, Fei Wang, Shunlong Wang, Zhiming Yuan, Han Xia.

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
