## [Decision Letter · Decision Letter 0]

20 Jul 2024

Dear Professor Xia,

Thank you very much for submitting your manuscript "Detection and evolutionary characterization of arboviruses in mosquitoes and biting midges on Hainan Island, China, 2019–2023" for consideration at PLOS Neglected Tropical Diseases. As with all papers reviewed by the journal, your manuscript was reviewed by members of the editorial board and by several independent reviewers. 

In light of the reviews (below this email), we would like to invite the resubmission of a significantly-revised version that should address the specific points made by me below and those made by each of the reviewers.

I believe that this is an important manuscript that documents the presence of some important viruses on Hainan Island. Below are a few additional comments that should be addressed.

1. Line 25 and many other places in the manuscript. Why report rates to two decimals of a percent? If one less or one more positive mosquito had been detected, the MIR would have changed by more than a full percent for many of the locations. This is reporting data to a level of accuracy not supported by the data and makes the reader question the accuracy of all of the data. Don’t report rates to more than one decimal place. See line 28 where if one less had been found, the MIR would have been 0, while if one more had been found, it would have been 14%, so a MIR of 7.142 is really meaningless.

2. Line 51: Was it 40,000 new cases or 40,000 deaths? I believe that there were about 96 million new cases.

3. Lines 185-187: You stated this correctly. Yes, you detected JEV RNA in 9 pools of mosquitoes, but only detected actual virus in three pools.

4. Line 195 (Table 1): As indicated in comment 3, you only detected JEV in 3 of the 9 JEV RNA detections. If JEV was not detected, isn’t it likely that only a small piece of noninfectious RNA was present in these other six pools? Should these unconfirmed positives but used to calculate a MIR? Similar comment for line 215 and BTV.

5. Lines 347-349: Why include the sentence on JE in the middle of the discussion of GETV? More importantly, the mere detection of GETV from a mosquito species does not mean that the species is a competent vector and that it is able to carry (i.e., transmit) that virus. However, I am fairly certain that it is a competent vector.

6. Lines 355-356: Was it the limited number of samples collected or was it the limited number of Ae. aegypti and Ae. albopictus collected?

7. Line 396+: References: These need to be formatted properly.

a. Only the first word and proper nouns in a reference title should be capitalized. See references 2, 6, 7, and many others. 

b. The first letter of each word in a journal title should be capitalized. See references 4, 14, 18, 20, and many others. 

c. Is it the “Chinese Journal of Vector Biology and Control” as in reference 3 or the “Chine Journal of Vector Biology and Control” as in reference 12?

d. Shouldn’t scientific names be in italics? See reference 14

e. Proper nouns, such as “Tembusu” should be capitalized. See reference 35 and others.

Minor Comments:

8. Line 66: Minor, but “…located in…zones” is a non-essential clause and should be separated by commas, i.e., “Island, located in…zones, is separated…”

9. Line 76: only proper nouns are capitalized in a virus name, so this should be chikungunya virus. Note, it was done correctly on line 57.

10. Line 124: What is RPMI? I know that it is Roswell Park Memorial Institute medium, but RPMI should have been established on line 95.

11. Line 125: There should be a space between “(PSA)” and “solution,” and why capitalize “solution?” See also line 128 where there needs to be a space between “-21” and “baby…”

12. Line 162 (Fig. 1): It might be nice to indicate that the clear areas were not sampled and even include their names.

13. Line 171: “Bluetongue” should be bluetongue, but more importantly, BTV was established on line 61.

14. Lines 251-253: This sentence is a bit confusing. Should, “…not only the RT-PCR detection, virus isolation, and deep sequencing were used in combination in this study.” have been, “…not only the RT-PCR detection, but virus isolation and deep sequencing were used in combination with RT-PCR in this study.”

15. Line 258: Shouldn’t “…isolate, but the…” be “…isolate, the full…” without “but?”

16. Lines 261-262: Shouldn’t “…mosquitoes Hainan…” be “…mosquitoes from Hainan…?”

17. Lines 269-270: Is it “G3” as on line 269 or “GIII” as on line 270?

18. Line 279: “…as the those…” should be “as those…” 

19. Line 280: Should “attenuation” be changed to “virulence?”

20. Line 309: Should “In here…” be changed to “In the current study,…?

21. Line 314: Should “and chickensTMUV, and…” (note the lack of a space after chickens) be changed to “and chickens, and…?”

22. Line 316: Why is “duck” capitalized?

We cannot make any decision about publication until we have seen the revised manuscript and your response to the reviewers' comments. Your revised manuscript is also likely to be sent to reviewers for further evaluation.

Sincerely,

Michael J Turell, Ph.D.

Academic Editor

Amy Morrison

Section Editor

I believe that this is an important manuscript that documents the presence of some important viruses on Hainan Island. Below are a few additional comments that should be addressed.

1. Line 25 and many other places in the manuscript. Why report rates to two decimals of a percent? If one less or one more positive mosquito had been detected, the MIR would have changed by more than a full percent for many of the locations. This is reporting data to a level of accuracy not supported by the data and makes the reader question the accuracy of all of the data. Don’t report rates to more than one decimal place. See line 28 where if one less had been found, the MIR would have been 0, while if one more had been found, it would have been 14%, so a MIR of 7.142 is really meaningless.

2. Line 51: Was it 40,000 new cases or 40,000 deaths? I believe that there were about 96 million new cases.

3. Lines 185-187: You stated this correctly. Yes, you detected JEV RNA in 9 pools of mosquitoes, but only detected actual virus in three pools.

4. Line 195 (Table 1): As indicated in comment 3, you only detected JEV in 3 of the 9 JEV RNA detections. If JEV was not detected, isn’t it likely that only a small piece of noninfectious RNA was present in these other six pools? Should these unconfirmed positives but used to calculate a MIR? Similar comment for line 215 and BTV.

5. Lines 347-349: Why include the sentence on JE in the middle of the discussion of GETV? More importantly, the mere detection of GETV from a mosquito species does not mean that the species is a competent vector and that it is able to carry (i.e., transmit) that virus. However, I am fairly certain that it is a competent vector.

6. Lines 355-356: Was it the limited number of samples collected or was it the limited number of Ae. aegypti and Ae. albopictus collected?

7. Line 396+: References: These need to be formatted properly.

a. Only the first word and proper nouns in a reference title should be capitalized. See references 2, 6, 7, and many others. 

b. The first letter of each word in a journal title should be capitalized. See references 4, 14, 18, 20, and many others. 

c. Is it the “Chinese Journal of Vector Biology and Control” as in reference 3 or the “Chine Journal of Vector Biology and Control” as in reference 12?

d. Shouldn’t scientific names be in italics? See reference 14

e. Proper nouns, such as “Tembusu” should be capitalized. See reference 35 and others.

Minor Comments:

8. Line 66: Minor, but “…located in…zones” is a non-essential clause and should be separated by commas, i.e., “Island, located in…zones, is separated…”

9. Line 76: only proper nouns are capitalized in a virus name, so this should be chikungunya virus. Note, it was done correctly on line 57.

10. Line 124: What is RPMI? I know that it is Roswell Park Memorial Institute medium, but RPMI should have been established on line 95.

11. Line 125: There should be a space between “(PSA)” and “solution,” and why capitalize “solution?” See also line 128 where there needs to be a space between “-21” and “baby…”

12. Line 162 (Fig. 1): It might be nice to indicate that the clear areas were not sampled and even include their names.

13. Line 171: “Bluetongue” should be bluetongue, but more importantly, BTV was established on line 61.

14. Lines 251-253: This sentence is a bit confusing. Should, “…not only the RT-PCR detection, virus isolation, and deep sequencing were used in combination in this study.” have been, “…not only the RT-PCR detection, but virus isolation and deep sequencing were used in combination with RT-PCR in this study.”

15. Line 258: Shouldn’t “…isolate, but the…” be “…isolate, the full…” without “but?”

16. Lines 261-262: Shouldn’t “…mosquitoes Hainan…” be “…mosquitoes from Hainan…?”

17. Lines 269-270: Is it “G3” as on line 269 or “GIII” as on line 270?

18. Line 279: “…as the those…” should be “as those…” 

19. Line 280: Should “attenuation” be changed to “virulence?”

20. Line 309: Should “In here…” be changed to “In the current study,…?

21. Line 314: Should “and chickensTMUV, and…” (note the lack of a space after chickens) be changed to “and chickens, and…?”

22. Line 316: Why is “duck” capitalized?

Reviewer's Responses to Questions

**Key Review Criteria Required for Acceptance?**

**Methods**

-Are the objectives of the study clearly articulated with a clear testable hypothesis stated?

-Is the study design appropriate to address the stated objectives?

-Is the population clearly described and appropriate for the hypothesis being tested?

-Is the sample size sufficient to ensure adequate power to address the hypothesis being tested?

-Were correct statistical analysis used to support conclusions?

-Are there concerns about ethical or regulatory requirements being met?

Reviewer #1: The study was comprised of a loose assemblage of field sampling using methods more or less specific for night-biting zoophagic species. The objectives for using this approach were not clearly described. This sampling was unexpected considering the recent outbreaks of dengue and chikungunya transmitted by anthropophagic day-biting Aedes that are best sampled by other methods. The sporadic nature of the sampling makes the MIR estimates collection specific and not a representation of overall risk. 

The handling of the collections in the field was not clearly indicated. Were the specimens frozen immediately after capture using dry ice or where the specimens transported alive to a regional laboratory. It was not clearly stated where the laboratory work and virus isolations were done.

The diagnostic and virological methods seem adequately presented and represent the 'heart' of the paper. However, there was no indication where the isolates were stored and whether the sequences were deposited in GenBank -- usually deposition references are included.

Reviewer #2: The methods used to generate the data presented in this manuscript are clearly presented and all of the information needed to reproduce the results is present.

Reviewer #3: The current manuscript lacks a clearly defined hypothesis and articulated aims. The authors should improve the abstract and introduction sections by including a specific scientific aim. Using epidemiological or historical data to frame this aim would enhance the study's context and relevance.

Furthermore, the manuscript would benefit from additional background information on the sampling collection process. Detailed information about the environments selected at each site, the criteria for site selection, the number of traps set per site, and the duration of the sampling effort in days should be included. It is important to specify whether the sampling effort was consistent across all collection sites.

Additionally, the authors should provide detailed information on the handling and transportation of the mosquitoes. Clarify whether the mosquitoes were identified in the field or the lab and describe how the cold chain was maintained during the identification process and transportation.

Please clarify whether you inoculate mosquito homogenates only from those with PCR-positive results or from all homogenates generated in the study.

**Results**

-Does the analysis presented match the analysis plan?

-Are the results clearly and completely presented?

-Are the figures (Tables, Images) of sufficient quality for clarity?

Reviewer #1: The results were clearly presented. It was not evident how many species were represented within the group labeled 'Culex sp.' nor was it explained why these mosquito specimens were not identified to species.

Reviewer #2: The results are mostly clearly presented. Figures and tables are good, although the resolution in the map figure made it hard to read (hard, but not impossible).

Reviewer #3: The authors could provide a more detailed description of the most important mosquito species and midges, focusing not only on general characteristics but also specifically in relation to the sampling collection. They do not need to describe all species per site, just the most frequent or medically important ones. Additionally, include the 95% binomial confidence interval in the results for mosquitoes positive for each virus, as it is important to convey the uncertainty around the central estimates.

**Conclusions**

-Are the conclusions supported by the data presented?

-Are the limitations of analysis clearly described?

-Do the authors discuss how these data can be helpful to advance our understanding of the topic under study?

-Is public health relevance addressed?

Reviewer #1: The authors' conclusions were well presented and the public/veterinary importance clearly indicated. A case could be made for using a One Health approach to monitoring arbovirus activity on Hainan in a more consistent manner to determine outbreak risk.

Reviewer #2: The conclusions are justified.

Reviewer #3: The authors are required to rewrite the discussion section and make adjustments once a clear aim is stated. They need to better discuss how their observations align with the current epidemiology in Hainan Island. If these viruses are not currently being detected in the region, the authors should explore possible reasons for this discrepancy. Based on their results, the authors should provide suggestions for improving laboratory and epidemiological surveillance in the region.

**Editorial and Data Presentation Modifications?**

Reviewer #1: Although well-prepared, I have recommended 'minor' revision to improve the presentation [see attached file for editorial suggestions and minor comments].

Reviewer #2: Minor edits noted:

Line 63: Sever should be severe.

Line 64: animal disease should be animal diseases.

Line 66: commas after the words island and zones.

Resoluion on figure 1 is poor.

Lines 167 - 171: This sentence is confusing. Consider rewording so that it's clear the total number of PCR positive pools. This is minor, but I had to read the sentence three times to understand it.

Table 1. Column heading should read 'habitat,' not 'habitant.'

Reviewer #3: The figures clearly match the results, and no changes are required at this stage.

**Summary and General Comments**

Reviewer #1: As mentioned earlier, the main weakness of the study is lack of consistency of sampling in time and space. The collection data are therefore confounded and can't really used to present clearly defined estimates of arbovirus risk. The data do indicate the presence and continued persistence of some zoonotic viruses known to be active in mainland China. Despite this, the data are unique for Hainan, a very different part of China.

Reviewer #2: Wu and colleagues present a study of mosquito- and midge-borne arboviruses in Hainan Island China. This is a surveillance-based study: arthropods were captured, identified, pooled and screened for viral RNA. Pools containing viral RNA were inocluated onto cells and viruses isolated. Nucleotide sequences were determined and phylogenetic analyses conducted. In addition, the mutations that differ from previously circulating viruses, and those that may confer interesting phenotypes are described. 

The manuscript reports isolation of several viruses, some are first records for this island. Thus, there is new information presented in this manuscript, although much of what is reported is confirmatory - we already knew that JEV was present on the island, for example.

Moreover, this is a straightforward and well done manuscript that will be of interest to readers of PNTD. My sole somewhat significant critique of the work is that I don't think that time and space have been well integrated into the analyses that are presented. Upon first reading, it was striking that the study ran from 2019 to 2023, but the only positive virus detections were recorded in 2022 and 2023. I had to dig into the supplement to learn that only a few pools were collected in 2019 and 2021, and none were collected in 2020. The lack of consistent sampling effort should be noted in the main text so that readers understand that the data collected may not be well suited to distinguishing fluctuations in epidemic or zoonotic risk during the period of observation. It would be helpful if the authors could comment on this in the discussion - they do a nice job of covering the genetic data, but the epidemiological data is handled less well.

Reviewer #3: The study presents a comprehensive survey of arboviruses in mosquitoes and biting midges on Hainan Island, China, conducted between 2019 and 2023. Utilizing RT-PCR assays, the researchers aimed to detect the presence, distribution, and infection rates of mosquito-borne pathogens. The study also employed cell inoculation and high-throughput sequencing to isolate viruses and assemble full viral genomes, followed by phylogenetic analysis to determine viral genotypes and evolutionary relationships.

The survey collected a substantial number of specimens, including 32,632 mosquitoes and 21,000 biting midges from 14 out of 18 cities/counties on the island. Notable findings include the detection of Japanese encephalitis virus (JEV), Tembusu virus (TMUV), Getah virus (GETV), Oya virus, and Bluetongue virus (BTV). The study successfully isolated and sequenced several virus strains, contributing valuable data to the understanding of arbovirus circulation on Hainan Island.

Specific comments and suggestions:

Introduce a clear scientific aim and hypothesis.

Include detailed descriptions of the most frequent or medically important mosquito species and midges at the sampling collection in the result section.

Provide comprehensive background information on the sampling methodology and environment.

Detail the handling and transportation procedures for the specimens in the methodology section.

Incorporate the 95% binomial confidence intervals in the results section.

Revise the discussion to align observations with current epidemiological data and suggest improvements for surveillance based on findings.

PLOS authors have the option to publish the peer review history of their article (what does this mean?). If published, this will include your full peer review and any attached files.

Reviewer #1: No

Reviewer #2: No

Reviewer #3: No
---

## [Editor Report · Decision Letter 1]

7 Oct 2024

Dear Professor Xia,

Thank you very much for submitting the revised version of your manuscript "Detection and evolutionary characterization of arboviruses in mosquitoes and biting midges on Hainan Island, China, 2019–2023" for consideration at PLOS Neglected Tropical Diseases. Thank you for addressing the reviewers’ and my comments, and the manuscript is much improved. I understand how difficult it can be for a person for whom English is not their primary language to write a manuscript in English. Overall, you did a great job, but I have suggested a few word changes. I have also indicated some additional questions/comments below. We are likely to accept this manuscript for publication, providing that you modify the manuscript according to the recommendations below.

1a. Lines 27 and 34 (and many others): Should numbers less than 10 be written out (as on line 34) or should numerals be used (as on line 27)? For PLoS NTD, I believe that either is acceptable, but you need to be consistent. Note, units of measurement are always numerals, i.e., 3 days. Because you have written out nearly all of the numbers <10 (which is the method that I prefer), I suggest that you go through the manuscript and write out all non-measurement numbers <10.

2. Line 98: I am not sure what you mean by “through cold chain.” Were the specimens maintained on dry ice, liquid nitrogen, kept frozen using regular ice, maintained in a refrigerated container, etc.? There are differences for virus isolation. Please be a little more specific.

3. Lines 160-164: The sum of these percentages is 93.3. Obviously, some other species (e.g., Ae. cinereus, Ae. vexans, Anopheles spp., etc.) were collected in limited numbers. You should add, “and other species in limited numbers (6.7%)” just before “(Fig.1 and S2 Table).”

4. Lines 214-224: What about the other six strains where the actual virus was not isolated? You mentioned that the entire E segment was present. Did you sequence it?

5. Lines 218-220: Are you only referring to the “latter isolate,” i.e., the one from Qionghai, or are you referring to all three isolates being within GI? If it is only the latter one, it should be, “The latter isolate clustered…” and if it is all three, it should be, “These isolates clustered…” What about the other 6 JEV detected; were they also GI?

6. Line 290: By “this species” do you mean Culex tritaeniorhynchus and Culex gelidus, or just Culex tritaeniorhynchus. I assume just Culex tritaeniorhynchus, so you might want to change the “this species” to “Culex tritaeniorhynchus.”

7. Line 296: Here it says that GIII was replaced by GI in the last 20 years. However, if GIII was still present in 2008, that was only 16 years ago and only 11 years before the start of the current study. Should 20 be changed to 15?

8. Lines 374: Yes, your detection of these viruses in potential vectors indicates that the diseases caused by these viruses are likely present on Hainan Island. As I have often said, if you do not look, the disease is not present. You might want to add, “However, the lack of reported disease may be due to a lack of recognition, rather than due to the absence of disease.” after Hainan Island and before DENV…

Minor Comments:

9. Line 27: as GETV was detected in only 1 city, it should be “from one city/county with…” As you stated that Oya and BTV were detected in Wanning City, why not indicated that GETV was detected in Qionghai, i.e., “from Qionghai with…”

10. Lines 45-47: I might modify this sentence to, “Except for arboviruses such as JEV, GTEV, and OYAV, which have been previously reported circulating on Hainan Island, TMUV and BTV were detected for the first time during this survey.”

11. Lines 79-80: DENV was already established on line 62 and JEV on line 63.

12. Line 96: “in the” collection site should be “at the” collection site.

13. Line 104: “…in biosafety laboratory” should be “in a biosafety laboratory.”

14. Line 127: This should be, “…study were inoculated…” and there should be a “,” after the “cells” of the BHK-21 cells.

15. Line 160: Change “were comprising…” to “consisted of…” 

16. Line 181: Check the spacing around the “.” Just before “The…”

17. Line 184: The “fifteen mosquito…” should be “15 mosquito…”

18. Line 195: There needs to be a space between “(0-4.0)” and “per…”

19. Line 200: “Aedes” should be on italics.

20. Lines 204-205: The samples were not inoculated with the cells, the cells were inoculated with the samples. Should this be, “…were inoculated onto C6/36…?” Also, because you are writing out most number <10, this should be “nine viral…” and on the next line, it should be “one strain…”

21. Line 284: Change “warnings, not only the RT-PCR…” to “warnings, RT-PCR…,” i.e., remove the “not only the” as you used all three methods.

22. Line 289: “Contained” should be “contained” without the capital “C.” Also, as Cx. has been established, these should be Cx. tritaeniorhynchus and Cx gelidus” (note, they should be in italics, but this program does not allow me to use italics.

23. Line 323: There should be a “,” after “flavivirus” and before the and.

24. Line 351: There should be a space between “1964” and “[47]…” 

25. Line 356: Shouldn’t “Hainan Provincial Center for Disease Control and Prevention…” be Hainan Provincial CDC…?”

26. Line 383-384: As “years” is a unit of measurement, it should be “3 years…” and “1 or 2 years.”

27. Line 384: It is not despite the inconsistencies that the data may not be well suited, it is because of the inconsistencies. I would modify the sentence to, “Because of the inconsistencies in…”

28. Line 463: Because “spp.” is not an actual species name, it should not be in italics.

29. Line 512: Culex tritaeniorhynchus should be in italics.

30. Line 528: Japanese Encephalitis should be Japanese encephalitis without the capital.

31. Line 587: Because “blue” is not a proper noun, it should not be capitalized.

Sincerely,

Michael J Turell, Ph.D.

Academic Editor

Amy Morrison

Section Editor

Figure Files:

Data Requirements:

Reproducibility:

References

---

## [Editor Report · Decision Letter 2]

20 Oct 2024

Dear Professor Xia,

We are pleased to inform you that your manuscript 'Detection and evolutionary characterization of arboviruses in mosquitoes and biting midges on Hainan Island, China, 2019–2023' has been provisionally accepted for publication in PLOS Neglected Tropical Diseases.

Thank you for responding to my previous comments/suggestions and submitting an improved manuscript. Despite a provision acceptance, I did find a few minor grammatical errors that need to be corrected. See below:

We are pleased to inform you that your manuscript 'Detection and evolutionary characterization of arboviruses in mosquitoes and biting midges on Hainan Island, China, 2019–2023' has been provisionally accepted for publication in PLOS Neglected Tropical Diseases.

Thank you for responding to my previous comments/suggestions and submitting an improved manuscript. Despite a provision acceptance, I did find a few minor grammatical errors that need to be corrected. See below:

1. Lines 61 and 62: “vectosr…” should “vectors…” and as Zika is a proper noun, unlike many arboviruses, it should be capitalized as it was correctly done in earlier versions of the manuscript.

2. Line 125: “( the number…” should be “(the number…” without the space before “the…”

3. Line 162: Because “Aedes albopictus” was used on line 129, this should be “Ae. albopictus…”

4. Lines 187-189: Why re-establish JEV and BTV again as they were established on lines 63 and 66?

5. Line 207: Insert a “,” after “BHK-21 cells” before the “and…”

6. Line 213: Check spacing before and after the “)” after “Cu-03…”

7. Line 363: There needs to be a “space” between “48]” and “All of…”

8. Line 367: I don’t think that you ever established “JE” as an abbreviation for Japanese encephalitis. You might want to do that on line77 or just write out Japanese encephalitis virus here and on line 368.

9. Line 405: I think that you forgot to include a “space after several commas, and a comma after Changjiang.

Best regards,

Michael J Turell, Ph.D.

Academic Editor

Amy Morrison

Section Editor

---

## [Editor Report · Acceptance letter]

25 Oct 2024

Dear Professor Xia,

We are delighted to inform you that your manuscript, "Detection and evolutionary characterization of arboviruses in mosquitoes and biting midges on Hainan Island, China, 2019–2023," has been formally accepted for publication in PLOS Neglected Tropical Diseases.

Best regards,

Shaden Kamhawi

co-Editor-in-Chief

Paul Brindley

co-Editor-in-Chief
